# Multi-institutional atlas of brain metastases informs spatial modeling for precision imaging and personalized therapy

Jorge Barrios[1,13], Evan Porter[1,13], Dante P. I. Capaldi[1], Taman Upadhaya[1], William C. Chen[1], Julian R. Perks[2], Aditya Apte[3,4], Michalis Aristophanous[3], Eve LoCastro[3,4], Dylan Hsu[4], Payton H. Stone[2], Javier E. Villanueva-Meyer[5,6], Gilmer Valdes[1,7,8], Fei Jiang[7], Michael Maddalena[9,10], Ase Ballangrud[3], Kayla Prezelski[3], Hui Lin[1,11], Jinger Y. Sun[2], Muhtada A. K. Aldin[2], Oi Wai Chau[1], Benjamin Ziemer[1], Maasa Seaberg[1], Penny K. Sneed[1], Jean L. Nakamura[1], Lauren C. Boreta[1], Shannon E. Fogh[1], David R. Raleigh[1,6,12], Jessica Chew[1], Harish Vasudevan[1,6], Soonmee Cha[5], Christopher Hess[5], Ruben Fragoso[2], David B. Shultz[10], Luke Pike[3], Shawn L. Hervey-Jumper[6], Derek S. Tsang[9], Philip Theodosopoulos[6], Daniel Cooke[5,6], Stanley H. Benedict[2], Ke Sheng[1], Jan Seuntjens[9,10], Catherine Coolens[9,10], Joseph O. Deasy[4], Steve Braunstein[1] & Olivier Morin[1,11] ✉

Brain metastases are a frequent and debilitating manifestation of advanced cancer. Here, we collect and analyze neuroimaging of 3,065 cancer patients with 13,067 brain metastases, representing an extensive collection for research. We find that metastases predominantly localize to high perfusion areas near the grey-white matter junction, but also identify notable differences depending on the primary cancer histology as well as brain regions which do not conform to this relationship. Lung and breast cancers, in contrast to melanoma, frequently metastasize to the cerebellum, hinting at biological pathways of spread. Additionally, the deep brain structures are relatively spared from metastasis, regardless of primary cancer type. Leveraging this data, we propose a probabilistic brain metastasis risk model to enhance the therapeutic ratio of whole-brain radiotherapy by targeting high risk areas while preserving cortical and subcortical brain regions of functional significance and low metastasis risk, potentially reducing the cognitive side effects of therapy.

An estimated 20% of cancer patients develop brain metastases (BM), and this devastating neurologic complication occurs with an annual incidence of over 200,000 new cases in the United States alone[1,2]. This prevalence is expected to rise in the years to come as therapeutic advances continue to improve extra-cranial metastasis control and prolong survival, and as detection increases via the widespread use of high-resolution imaging[3]. Numerous challenges contribute to poorer survival among cancer patients with BM, including poor penetration of

traditional chemotherapies into the central nervous system (CNS), limitations of existing local therapies for BM, such as surgery and radiotherapy, and the debilitating and often irreversible symptoms arising from intracranial progression. At the same time, understanding of the unique biology and microenvironment of CNS metastases has lagged that of their systemic counterparts, and our therapeutic resources, while growing, remain limited[4]. Despite efforts to pool and leverage large-scale imaging and health data to advance oncologic

care, even basic data regarding the distribution and morphology of BM across functional or vascular territories in the brain remain scarce. Such information would be useful for multiple reasons: First, biologically, the preference of BM for certain territories of the brain and the heterogeneity of these preferences across cancer types, may provide unique insights into the processes and prerequisites of BM development. Second, clinically, such information could guide the personalization of brain surveillance and CNS-directed therapies, particularly with respect to the design of function-sparing radiotherapy. Third, informatically, large-scale segmentation and quantitative evaluation of BM will enable future efforts to detect and longitudinally track these tumors, allowing the generation of predictive models of BM development, progression, and treatment response.

Initial efforts towards mapping the spatial distribution of BM used simplified anatomical atlases and manual image registration of limited patient cohorts and tumor burden[5,6]. Despite such limitations, these early studies revealed striking non-uniformity in location and predilection towards BM formation at the white-gray matter border as well as vascular watershed zones, leading to the hypothesis that BM arise via tumor microemboli which seed in regions of reduced blood flow[7–9]. Since then, elegant in vivo experiments have confirmed the key role of vascular flow in permitting the arrest and extravasation of circulating tumor cells and emboli[10]. Nevertheless, many questions remain regarding the necessary conditions for BM development, and modern efforts to map BM distribution in greater detail across functional and vascular territories may shed further light on the interplay between circulating tumor cells and the unique brain microenvironment.

An estimated 200,000 BM patients per year in the United States receive palliative whole-brain radiotherapy (WBRT), which remains a mainstay of therapy despite its well-known impacts on cognition, memory, and quality of life[11–15]. Although advances have been made in the use of stereotactic radiosurgery (SRS), a focused form of radiation associated with improved neuro-cognitive preservation, this treatment is usually reserved for patients with high-performance status and a limited number of small, well-defined lesions[2,3]. Novel CNS-penetrant systemic therapies have entered clinical use for certain cancers, but only a minority of BM patients are currently eligible[16,17], though this may change in the future. Hippocampal-avoidance WBRT (WBRT-HA) combined with memantine has recently been demonstrated to provide a modest improvement in cognitive outcomes, but patients still suffer significant neuro-cognitive decline[18]. It is possible that sparing of additional neuro-cognitive structures could result in further improvements, but a more detailed understanding of the risk of BM development across functional brain territories is necessary to guide these efforts. To this end, large-scale spatial mapping of BM represents a critical first step towards more radical personalized functional sparing approaches in radiotherapy design[19–23]. Associated with treatment, automated methods for tracking these lesions longitudinally will be necessary.

The advent of deep learning has greatly facilitated the automated detection, segmentation, and longitudinal evaluation of BM. However, the segmentation and detection performance of deep learning algorithms has recently plateaued below the threshold of clinical utility[24–27]. To achieve the high performance seen in other medical domains, BM databases that are far larger than those previously assembled are required[28]. To reach this scale, cooperative efforts across multiple institutions using a standardized data collection framework are needed, as well as novel methods to increase the effectiveness of limited training data, such as informed machine learning, a technique that leverages prior knowledge of lesion spatial distributions, morphology, and brain anatomy to constrain the solution space[29,30]. The development of this infrastructure will enable the future development and implementation of predictive models of BM growth and treatment response to better personalize clinical management and treatment recommendations.

In our investigation, we aim to thoroughly address a fundamental yet crucial inquiry: *where do metastases manifest within the brain?* We report on the creation of a scalable data science infrastructure to collect and standardize high-resolution magnetic resonance (MR) imaging and segmentation across multiple institutions, enabling the collection of a large database of BM to power necessary studies. We report on the spatial and morphological distributions of 13,067 BM from 3065 patients treated at four institutions, from which we have generated a detailed map of metastasis probability across anatomical, functional, and vascular territories of the brain. By combining anatomic and perfusion imaging, we demonstrate that the vast majority of BM occur within a narrow subspace defined by moderate to high perfusion and close proximity to the gray-white junction, and that the deep structures of the brain are relatively spared across all primary histologies. Additionally, we identify heterogeneity in the spatial distribution of BM arising from different primary cancers, as well as in the predilection of BM to form in different vascular territories. Leveraging this detailed map, we develop a probabilistic framework for the design of function-sparing brain radiotherapy, allowing for risk-based modulation of metastasis coverage and avoidance of neuro-cognitive structures.

## Results

### Data science infrastructure

In order to aggregate multi-institutional BM cases into a shareable resource, we created an open-source pipeline in the Python programming language and a scalable, robust, and efficient data science infrastructure to rapidly ingest (170 studies per second), standardize, and process MR imaging data with associated manual segmentations. A data entry template (provided here https://osf.io/fkqmr/) was shared between the participating institutions to standardize clinical data per patient. The integrated processing workflow facilitates data curation and ensures compatibility with a comprehensive structured query language (SQL) database[31], which serves as a repository for subsequent analysis (Fig. 1). Images are pre-processed, co-registered, and mapped to standardized Montreal Neurological Institute (MNI) anatomical, functional, and vascular brain atlases, allowing for the generation of BM risk maps. The infrastructure, code base, and database comprising 13,067 BM across 3065 MRIs from as many unique patients from four institutions (Table 1), are intended as a community resource in support of efforts to enhance auto-segmentation, detection, and quantitative analysis of BMs, and is easily extensible to incorporate data from additional institutions as well as longitudinal data on existing patients.

### Patient and brain metastasis characteristics

The cohort reported here includes patients presenting for first time SRS following a new diagnosis of BM at four academic medical centers in the United States and Canada between January 2011 and December 2023 (Table 1), comprising of 43.16% lung, 18.69% breast, 13.34% melanoma, 5.06% gastrointestinal, 2.68% genitourinary and 17.06% other primary cancer patients. Patients had a median of 2 metastases (range 1–53) with a median disease volume of 0.23 cc (range 0.002–62.381). Breast patients had the highest mean = 5/median = 3 lesion count combination (range 1–53), and gastrointestinal patients had the highest median disease volume of 0.494 cc (range 0.007–44.776). This study primarily focused on spatial BM risk analysis for lung, breast, and melanoma due to the statistical power provided by their natural prevalence in the overall cohort.

### Brain metastasis spatial distribution

The spatial distribution of BM by anatomical regions was non-uniform for lung, breast and melanoma primaries (Fig. 2A–C, Supplemental Fig. 1), and the majority of BMs (~72%) were present within 5 mm of the white-gray matter interface (Fig. 2D). Additional analysis demonstrated that the observed spatial distribution of BMs is

significantly closer ([with a 10% over-representation, *P* value < 0.05], Wilcoxon-signed rank test) to the white-gray matter interface than expected under the null hypothesis of brain "fjords-liked" convoluted morphology. These results indicate that the proximity of BMs to the white-gray matter interface reflects a meaningful spatial association rather than an artifact of brain morphology (Supplemental Fig. 2). As expected in a dataset of patients undergoing radiosurgery, most lesions had a high sphericity index (>0.75) and a volume smaller than 0.5 cc (Fig. 2E). The broad sphericity range (0.6–0.9) observed for volumes below 0.4 cc is attributed to the clinical practice of delineating a "cap" on lesions in the superior-inferior direction. This approach notably reduces sphericity scores, especially in older scans acquired with larger slice thicknesses. Regions of the brain with the greatest cumulative burden of BMs were the frontal and parietal lobes and cerebellum (Fig. 2B, Supplemental Figs. 1 and 3, Supplemental Table 1), owing partly to the relatively large volume of these regions (35.7%, 18.6%, and 8.6%, respectively, of brain volume). The relative contribution by region remained consistent when applying 5- or 10-mm expansions as a sensitivity analysis (Supplemental Fig. 3), indicating that mapping misregistration in MNI space and borderline metastases did not have an outsized effect on these findings. Furthermore, there was no significant difference in BM distribution by laterality (Supplemental Fig. 4) of the anatomical regions in the atlas.

When normalized by brain volume (Fig. 2C), it was apparent that there was a significant (2.8-times, 4-times) enrichment of BMs within the cerebellum for lung and breast cancers, respectively, as compared to melanoma ([Melanoma-Breast *P* value = 0.0018, Melanoma-Lung *P* value = 0.0001], Welch's *t* test). Interestingly, there was also less dramatic but statistically significant heterogeneity in BM density within the temporal lobe for lung and breast cancers as compared to melanoma ([Melanoma-Breast *P* value = 0.011, Melanoma-Lung *P* value = 0.036], Welch's *t* test), while no significant difference was found between breast cancer and lung cancer (*P* value = 0.153, Welch's *t* test). The regions with the lowest absolute risk of BM were mainly comprised of subcortical brain structures such as the brainstem, thalamus, hippocampi, pituitary, ventricles, and amygdala. Collectively, these structures contained just ~6–8% of BM. Examining distribution by finer anatomical regions, additional notable regions were identified with relatively decreased BM density and a low contribution (<1–2%) to BM burden, including the cingulate (0.25% of metastases, density rank 39 out of 50, Supplemental Table 1), and orbitofrontal cortex (0.33% of metastases, density rank 35 out of 50). Conversely, certain sub-regions appeared to be preferentially enriched in BM density, particularly the pre-central/post-central gyri (4.1% of metastases, density rank 2 out of 50), pre-cuneus/cuneus (2.6% of metastases, density rank 5 out of 50), as well as the caudal middle frontal lobe (2.2% of metastases, density rank 6 out of 50) and superior parietal lobe (2.6% of metastases, density rank 4 out of 50).

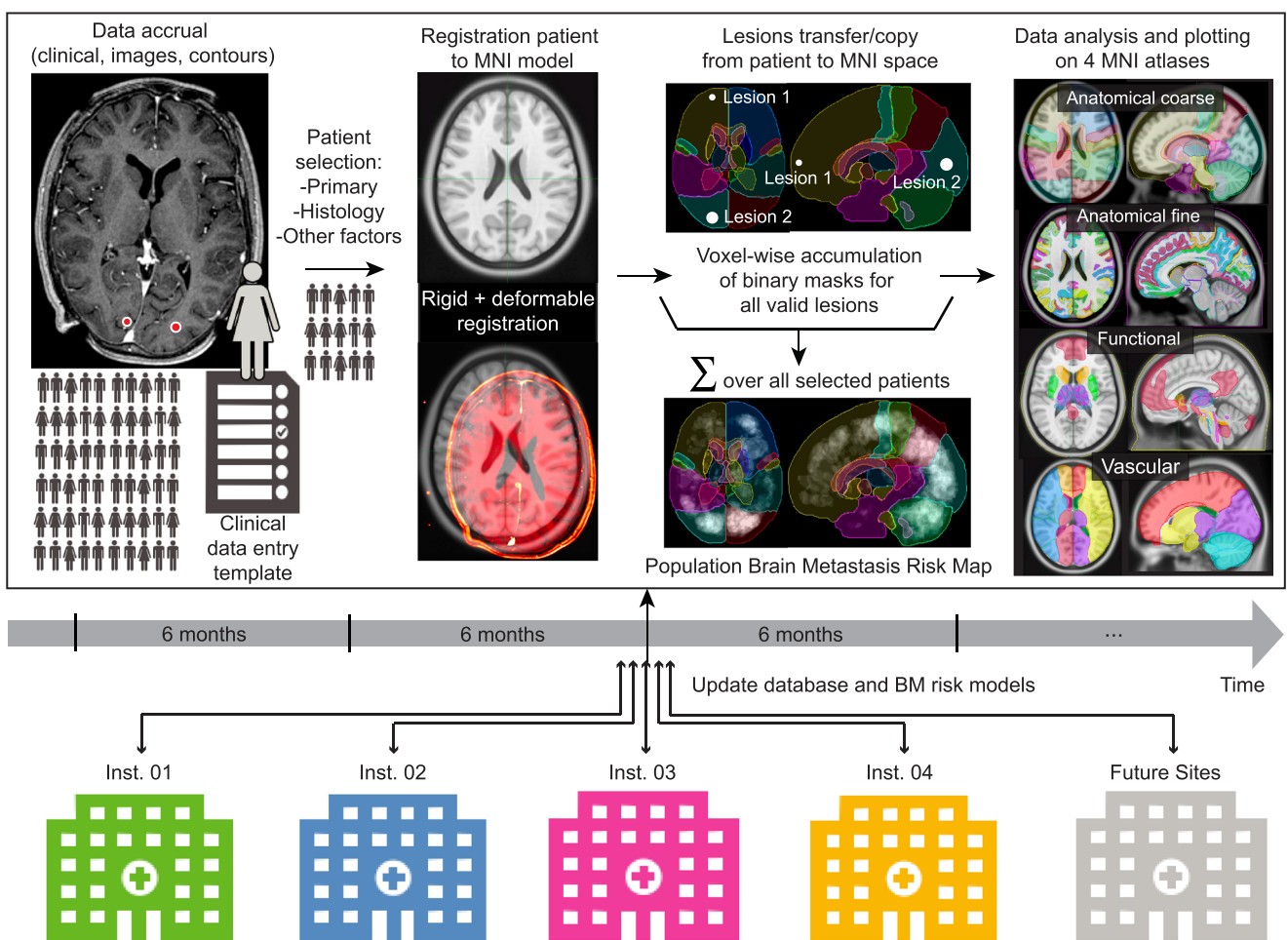

**Fig. 1 | Multi-institutional methodology for building spatial risk maps of brain metastasis.** T1-weighted MR post-gadolinium contrast images and a clinical data acquisition template are used at all partner institutions to accumulate the raw information. Patient images co-registered (rigid + deformable) to the MNI model and brain metastases transferred and accumulated into the MNI space. The BM risk map in MNI space is plotted against four validated atlases: anatomical coarse, anatomical fine, functional, and vascular. The data processing pipeline and transfer is designed to be updated every 6 months with new data from current and future participating institutions.

**Table 1 | Demographic, clinical, and brain metastasis characteristics split by the primary cancer for the data accumulated at the four partner institutions**

| Characteristic | MSKCC (n = 510) | PMH (n = 518) | UCD (n = 488) | UCSF (n = 1549) | Total (n = 3065) |
|---|---|---|---|---|---|
| **Lung** | | | | | |
| Patients, n | 222 | 253 | 248 | 600 | 1323 |
| Lesions, n | 664 | 1016 | 655 | 2989 | 5324 |
| Age, mean (range) | 65 (33–93) (n = 201) | 66 (33–89) (n = 151) | 65 (29–91) | 64 (32–91) | 65 (29–93) (n = 1200) |
| Sex, n (M/F) | 106/116 | 114/139 | 120/128 | 274/326 | 614/709 |
| Lesions per patient, median/mean (range) | 2/2.99 (1–17) | 2/4.02 (1–21) | 2/2.64 (1–15) | 3/4.98 (1–48) | 2/4.02 (1–48) |
| Disease volume, mean (range) (cc) | 1.561 (0.032–25.541) | 0.893 (0.009–43.210) | 1.831 (0.007–40.484) | 1.108 (0.006–53.620) | 1.213 (0.006–53.620) |
| Sphericity, mean (range) | 0.849 (0.612–0.916) | 0.834 (0.545–0.922) | 0.831 (0.544–0.912) | 0.814 (0.561–0.948) | 0.827 (0.544–0.948) |
| Histology top 3, n (%) | Adenocarcinoma 209 (94.1) Squamous Cell 8 (3.6) Other 3 (1.4) | N/A | Adenocarcinoma 138 (55.6) Other 88 (35.5) Squamous Cell 20 (8.1) | Adenocarcinoma 473 (78.8) Small Cell 31 (5.2) Carcinoma NOS 31 (5.2) | Adenocarcinoma 980 (76.6) Other 91 (8.5) Squamous Cell 59 (5.5) |
| **Breast** | | | | | |
| Patients, n | 87 | 64 | 74 | 348 | 573 |
| Lesions, n | 382 | 309 | 267 | 2120 | 3078 |
| Age, mean (range) | 54 (30–81) (n = 81) | 58 (33–83) (n = 40) | 56 (30–87) | 53 (0–79) | 54 (30–87) (n = 543) |
| Sex, n (M/F) | 0/87 | 1/63 | 0/74 | 5/343 | 6/567 |
| Lesions per patient, median/mean (range) | 2/4.39 (1–18) | 4/4.83 (1–19) | 2/3.61 (1–18) | 4/6.09 (1–44) | 3/5.37 (1–44) |
| Disease volume, mean (range) (cc) | 1.581 (0.019–62.381) | 0.913 (0.008–21.640) | 1.259 (0.013–16.343) | 1.064 (0.002–48.750) | 1.151 (0.002–62.381) |
| Sphericity, mean (range) | 0.833 (0.531–0.914) | 0.827 (0.529–0.916) | 0.816 (0.504–0.899) | 0.805 (0.444–0.922) | 0.812 (0.444–0.922) |
| Histology top 3, n (%) | Adenocarcinoma 84 (96.6) Other 2 (2.3) Unknown 1 (1.1) | N/A | Other 65 (87.8) Adenocarcinoma 5 (6.8) Unknown 4 (5.4) | Adenocarcinoma 337 (96.8) Other 8 (2.3) Bone 1 (0.3) | Adenocarcinoma 426 (83.7) Other 75 (14.7) Unknown 5 (1.0) |
| **Melanoma** | | | | | |
| Patients, n | 62 | 56 | 55 | 236 | 409 |
| Lesions, n | 249 | 287 | 148 | 1312 | 1996 |
| Age, mean (range) | 63 (20–91) (n = 56) | 61 (27–85) (n = 31) | 59 (23–81) | 61 (29–101) | 61 (20–101) (n = 378) |
| Sex, n (M/F) | 40/22 | 40/16 | 37/18 | 147/89 | 264/145 |
| Lesions per patient, median/mean (range) | 2/4.02 (1–21) | 3/5.12 (1–27) | 2/2.69 (1–12) | 3/5.56 (1–51) | 3/4.88 (1–51) |
| Disease volume, mean (range) (cc) | 1.456 (0.030–38.727) | 1.362 (0.005–52.918) | 2.385 (0.005–14.842) | 1.326 (0.004–33.036) | 1.493 (0.004–52.918) |
| Sphericity, mean (range) | 0.847 (0.683–0.910) | 0.837 (0.654–0.906) | 0.824 (0.430–0.909) | 0.819 (0.502–0.922) | 0.825 (0.430–0.922) |
| Histology top 3, n (%) | Melanoma 62 (100) | Melanoma 56 (100) | Melanoma 55 (100) | Melanoma 236 (100) | Melanoma 409 (100) |
| **Gastrointestinal** | | | | | |
| Patients, n | 31 | 15 | 19 | 90 | 155 |
| Lesions, n | 87 | 74 | 39 | 287 | 487 |
| Age, mean (range) | 62 (34–87) (n = 27) | 68 (54–85) (n = 6) | 58 (35–77) | 58 (24–81) | 59 (24–87) (n = 142) |
| Sex, n (M/F) | 17/14 | 8/7 | 12/7 | 52/38 | 89/66 |
| Lesions per patient, median/mean (range) | 1/2.81 (1–13) | 3/4.93 (1–14) | 1/2.05 (1–7) | 2/3.19 (1–14) | 2/3.14 (1–14) |
| Disease Volume, mean (range) (cc) | 2.576 (0.046–16.416) | 1.554 (0.007–16.607) | 4.922 (0.036–39.471) | 2.990 (0.016–44.776) | 2.853 (0.007–44.776) |
| Specificity, mean (range) | 0.840 (0.602–0.913) | 0.825 (0.577–0.919) | 0.831 (0.669–0.897) | 0.825 (0.645–0.919) | 0.829 (0.577–0.919) |
| Histology top 3, n (%) | Adenocarcinoma 31 (100) | N/A | Adenocarcinoma 14 (73.7) Other 4 (21.1) Squamous Cell 1 (5.3) | Adenocarcinoma 67 (74.4) Carcinoma NOS 12 (13.3) Squamous Cell 7 (7.8) | Adenocarcinoma 112 (80) Other 12 (8.6) Squamous Cell 8 (5.7) |
| **Genitourinary** | | | | | |
| Patients, n | 12 | 15 | 25 | 30 | 82 |
| Lesions, n | 35 | 71 | 75 | 140 | 321 |
| Age, mean (range) | 59 (23–82) | 62 (43–82) (n = 11) | 62 (21–84) | 54 (20–78) | 58 (20–84) (n = 78) |
| Sex, n (M/F) | 8/4 | 4/11 | 10/15 | 27/3 | 49/32 |

**Table 1 (continued) | Demographic, clinical, and brain metastasis characteristics split by the primary cancer for the data accumulated at the four partner institutions**

| Characteristic | MSKCC (n = 510) | PMH (n = 518) | UCD (n = 488) | UCSF (n = 1549) | Total (n = 3065) |
|---|---|---|---|---|---|
| Lesions per patient, median/mean (range) | 2/2.92 (1–10) | 3/4.73 (1–14) | 2/3.0 (1–11) | 2/4.67 (1–26) | 2/3.91 (1–26) |
| Disease volume, mean (range) (cc) | 2.224 (0.109–18.725) | 1.407 (0.016–14.684) | 2.225 (0.008–23.941) | 3.022 (0.016–52.342) | 2.401 (0.008–52.342) |
| Sphericity, mean (range) | 0.859 (0.756–0.904) | 0.827 (0.642–0.907) | 0.839 (0.655–0.911) | 0.820 (0.614–0.904) | 0.833 (0.614–0.911) |
| Histology top 3, n (%) | Adenocarcinoma 9 (75) Other 3 (25) | N/A | Other 13 (52) Adenocarcinoma 9 (36) Squamous Cell 2 (8) | Carcinoma NOS 12 (40) Germ Cell Tumor 8 (26.7) Other 4 (13.3) | Other 20 (29.9) Adenocarcinoma 18 (26.9) Carcinoma NOS 12 (17.9) |
| Other | | | | | |
| Patients, n | 96 | 115 | 67 | 245 | 523 |
| Lesions, n | 302 | 349 | 129 | 1081 | 1861 |
| Age, mean (range) | 61 (19–88) (n = 86) | 60 (24–81) (n = 39) | 61 (16–90) | 58 (4–88) | 59 (4–90) (n = 437) |
| Sex, n (M/F) | 43/53 | 47/27 (n = 74) | 34/33 | 132/113 | 256/226 (n = 482) |
| Lesions per patient, median/mean (range) | 2/3.15 (1–19) | 2/3.03 (1–18) | 1/1.93 (1–8) | 3/4.41 (1–32) | 2/3.56 (1–32) |
| Disease volume, mean (range) (cc) | 1.549 (0.017–36.346) | 1.269 (0.010–25.019) | 2.213 (0.037–14.708) | 1.638 (0.011–58.960) | 1.594 (0.010–58.960) |
| Sphericity, mean (range) | 0.854 (0.676–0.911) | 0.835 (0.596–0.911) | 0.811 (0.436–0.905) | 0.816 (0.544–0.911) | 0.825 (0.436–0.911) |
| Histology top 3, n (%) | Adenocarcinoma 42 (43.8) Renal cell 30 (31.2) Other 20 (20.8) | N/A | Other 32 (47.8) Renal Cell 16 (23.9) Adenocarcinoma 12 (17.9) | Renal Cell 69 (28.2) Carcinoma NOS 46 (18.8) Adenocarcinoma 40 (16.3) | Renal Cell 115 (28.2) Adenocarcinoma 94 (23.0) Other 57 (14.0) |

NOS not otherwise specified.

## Brain metastasis distribution by functional regions

Next, we sought to examine the distribution of BMs in the context of previously defined functional regions and networks of the brain. As subcortical structures appeared to be relatively spared from BMs and therefore an attractive target for functional sparing, we chose to examine four atlases delineating functional connectivity between subcortical structures related to cognitive function: (1) the Reich 2022[19] cognitive decline network atlas examined subcortical networks connected to regions targeted by deep brain stimulation in patients who experienced cognitive decline from this treatment; (2) the Tang 2017[22] brainstem connectome atlas utilized high-resolution imaging from the Human Connectome Project to delineate key fiber tracts within the brainstem and cerebellum; (3) the He 2020[20] basal ganglia and thalamus atlas utilized functional MRI data to delineate thalamocortical and basal ganglia-thalamic networks; and 4) the Behrens Oxford thalamic connectivity atlas[23] is a classic atlas of connections between the thalamus and cortical regions (Fig. 3A–B). We found that BM contribution within functional networks with greater cortical components (Reich 2022) largely correlated with the overall volume of the respective functional network and had similar BM density compared to other functional brain regions such as primary sensory cortex (Fig. 3C). Interestingly, the primary motor cortex was found to be enriched for BM compared to primary sensory cortex for lung cancer primary (P value = 0.0002, Welch's t test). As a proof of concept, we examined the percent of metastases captured while sparing/excluding successively greater anatomic or functional regions (Fig. 3D), ranging from 100% coverage with traditional WBRT, 99.4% coverage with traditional WBRT-HA, to 93.9% coverage with sparing of key subcortical structures, which we name WBRT-PROTECT (Personalized Radiation Optimization To Eliminate Collateral Toxicity), and 83.8% coverage with theoretical sparing of all functional regions identified from the Reich and He atlases.

## Brain metastasis distribution by arterial territories and level of perfusion

Next, given the non-uniformity of BM distribution, we sought to examine the occurrence of BMs across arterial territories and in relation to vascular perfusion as illustrated in Fig. 4A. We observed variation in BM burden (Fig. 4B) and density (Fig. 4C) by arterial territory. Melanoma metastases were more frequently located in regions supplied by the anterior circulation (74.4%, versus 55.8% in breast and 64.2% in lung). When mapped to a normalized perfusion map of the brain, we found that the majority of BM volume (~90%) resided within a subspace defined by moderate to high perfusion (Fig. 4D, 0.6–0.95 normalized perfusion), and within 5 mm of the white-gray matter interface (Fig. 4E). The prevalence of high BM density also sharply decreased below a normalized perfusion of approximately 0.6 (Fig. 4D). For lung and breast primaries, the inferior portion of the cerebellum was found to have statistically more BM compared to the superior portion (P values < 0.01, Welch's t test) due to its larger volume, however the superior portion has a higher BM density (P values < 0.001, Welch's t test). When examined by anatomical region, there was a trend towards increased BM density with increased perfusion (Supplemental Fig. 5, Supplemental Fig. 6). Examination of these anatomical regions identified outliers of note: in lung and breast cancers, the cerebellar white matter and cerebellar cortex had among the highest brain metastasis density despite a moderate perfusion (ratio 2.5–2.75). Conversely, the regions of the cingulate, including the caudal anterior, posterior, rostral anterior, and isthmus, had a low BM density despite moderate to high perfusion (ratio 0.5–0.75), across all three primary cancers.

## Function sparing and personalized brain radiotherapy for brain metastases

Finally, incorporating our observations regarding the relative sparing of subcortical structures from BMs, we designed representative versions of WBRT reflecting various levels of functional sparing and expected coverage percentage of undetectable or future disease (Fig. 5). WBRT-PROTECT was able to reduce the radiation dose delivered to hippocampi, amygdala, thalamus, basal ganglia, and brainstem, by ~66% using photon-based intensity-modulated radiotherapy, and by ~90% using a proton-based plan (Fig. 5, Supplemental Fig. 7) while keeping the expected BM coverage at ~94% (Fig. 3D).

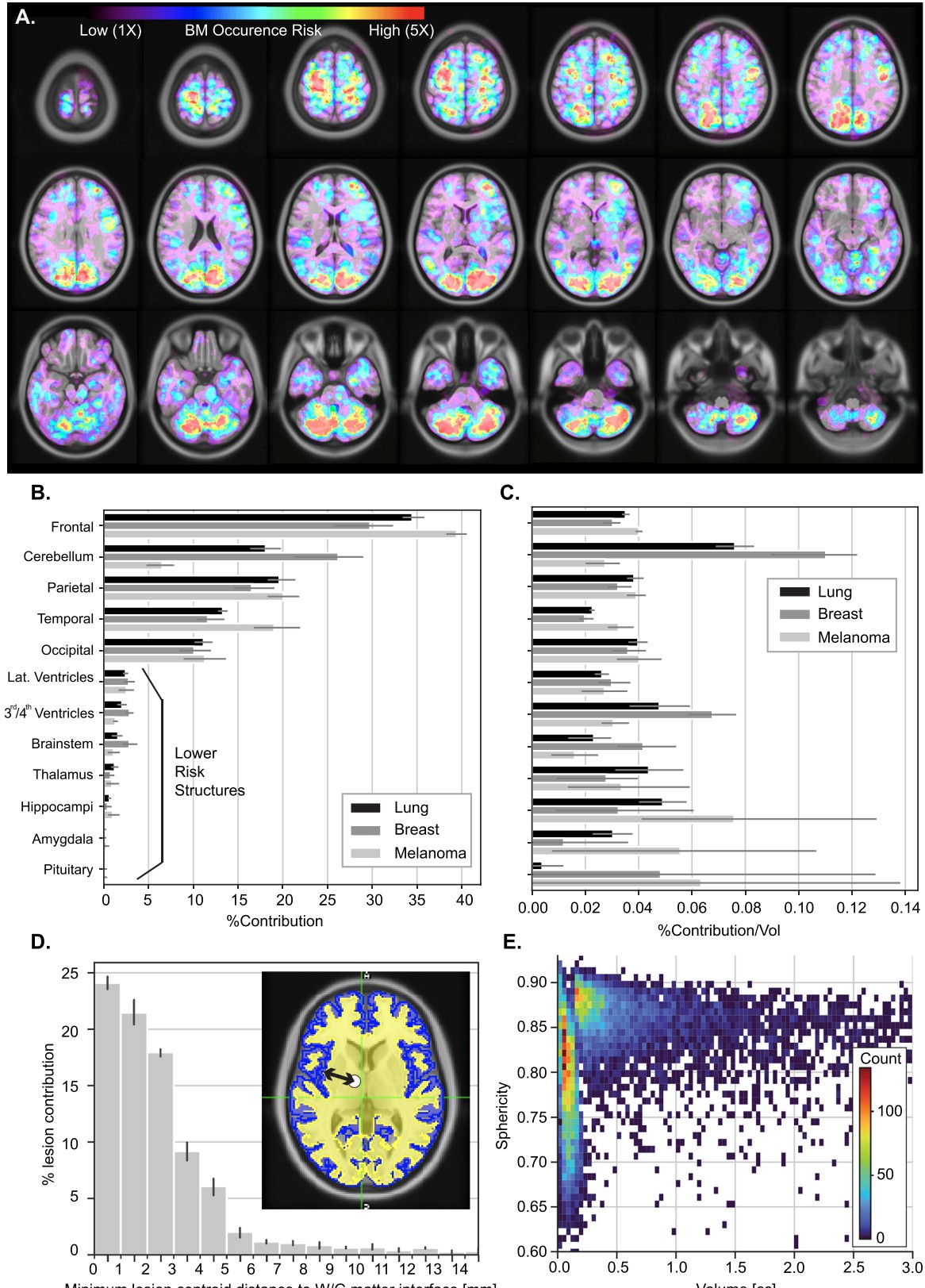

WBRT-PROTECT would spare cortical and subcortical brain regions (~5.5% of the brain volume) of documented and suspected functional significance. Complete dose-volume histograms comparisons of the conventional WBRT techniques to the newly proposed WBRT-

PROTECT (Supplemental Figs. 7 and 8) show significant dose sparing could be achieved. Supplemental Fig. 9 shows possible treatment approaches of WBRT-PROTECT that would differ based on the primary cancer histology, which demonstrated unique spatial BM risk.

**Fig. 2 | Brain metastasis spatial distribution and morphology by coarse anatomical regions. A** Brain metastasis risk levels (color-coded localized incidence risk of brain metastases) displayed on axial slices of the MNI model (grayscale) for patients with lung primary (*N* = 1323 patients, *L* = 5334 lesions) cancer, first diagnosis of brain lesions prior to stereotactic treatment. **B** Percentage contribution of BM lesions by coarse anatomical regions for lung (*N* = 1315 patients, *L* = 5230 lesions), breast (*N* = 570 patients, *L* = 3007 lesions), and melanoma (*N* = 407 patients, *L* = 1931 lesions) primary cancer. The cumulative lesion contribution of all brain regions represents 100%. Error bars indicate 95% confidence interval over the four institutions represented in the dataset. The center of the error bars represents the mean percentage contribution of all four institutions for a given structure. **C** Percentage contribution of brain metastasis normalized by the volume by coarse anatomical regions for lung (*N* = 1315 patients, *L* = 5230 lesions), breast (*N* = 570 patients, *L* = 3007 lesions), and melanoma (*N* = 407 patients, *L* = 1931 lesions)

primary cancer. Error bars indicate 95% confidence interval over the four institutions represented in the dataset. The center of the error bars represents the mean percentage contribution normalized by the volume of all four institutions for a given structure. **D** Percentage of the brain lesion contribution as a function of the minimum distance of the lesion centroid to the white and gray matter interface. Error bars indicate 95% confidence interval over the four institutions represented in the dataset (*N* = 2113 patients, *L* = 8176 lesions). The center of the error bars represents the mean percentage contribution of all four institutions for a specific distance to the white-gray matter junction. The white and gray matter volumes are represented by yellow and light blue, respectively. The dark blue represents the edge of the brain's gray matter. **E** Lesions sphericity distribution as a function of the lesion volume (*N* = 2001 patients, *L* = 9302 lesions), where the heatmap represents the magnitude of the count in the plotted joint histogram distribution. Source data are provided as a Source Data file.

## Discussion

Here, we report the largest multi-institutional brain metastasis database to date, from which we develop a detailed map of BM probability across anatomic, functional, and vascular territories of the brain. In line with prior studies, we find striking non-uniformity in the spatial distribution of BM. Combining anatomic and perfusion imaging, we find that the majority of BM, ~72%, developed within 5 mm of the gray–white matter junction and in regions of moderate to high normalized perfusion level, with relative sparing of the deep structures of the brain such as the thalami, hippocampi, amygdalae, internal capsule, and brainstem. Examination of vascular territories and perfusion maps identified a general association between perfusion and BM density, though some regions, such as the orbitofrontal cortex and cingulate cortex, appeared to deviate from this relationship, with relatively fewer metastases despite moderate to high perfusion. These deviations may reflect underlying biological differences in tumor cell tropism, local vascular architecture, or unique microenvironmental factors such as blood–brain barrier integrity, immune response, or neuronal interactions that would need further study. At the same time, we identify heterogeneity in the distribution of BM between primary cancer types, finding the cerebellum to be over-represented as a metastasis site for lung and breast cancers, but relatively under-represented for melanomas. Specifically, melanoma metastases are more frequently located in the anterior circulation, whereas lung and breast cancers demonstrate a higher prevalence in the posterior circulation, particularly in the cerebellum. This suggests potential differences in tumor cell dissemination and seeding within vascular territories. However, we acknowledge that providing a definitive mechanistic explanation for these differences is beyond the capabilities of this study. Finally, applying these observations, we develop a probabilistic framework for the design of function-sparing palliative radiotherapy, allowing for titration of metastasis coverage and avoidance of anatomical or functional neuro-cognitive structures.

Our findings corroborate historical studies and clinical experience, which have demonstrated the non-uniform distribution of BM, favoring the gray-white matter junction and vascular watershed regions. Although regions of moderate to high perfusion predominate overall, we identify regions in which this relationship is mismatched, namely, subcortical structures, where metastases are rare, despite relatively moderate to high perfusion. Differences in the relationship between perfusion and metastasis risk between evolutionarily distinct regions of the brain have been postulated before[32], and our findings might suggest differences in vascular and microvasculature structures in these regions to underlie reduced metastasis formation, although other factors such as differences in the blood–brain barrier, tissue architecture, immune and neuronal microenvironment, remain uninvestigated. The observed differences in the perfusion-BM relationship across cancer types (Supplemental Fig. 6) are intriguing and may reflect underlying biological and vascular differences between tumor types. One

potential explanation is that breast cancer metastases may exhibit a greater reliance on alternative mechanisms of dissemination and colonization beyond perfusion-driven seeding. For example, prior studies suggest that breast cancer brain metastases have unique interactions with the blood–brain barrier and perivascular niche, potentially involving distinct adhesion molecules, immune microenvironment factors, or extravasation pathways that differ from lung cancer and melanoma[33]. Additionally, breast cancer subtypes (e.g., HER2-positive, triple-negative) may exhibit heterogeneous perfusion dependencies, which could contribute to the less consistent perfusion-BM relationship observed in our analysis[34–36]. One notable finding we report is a substantial difference in risk of cerebellar metastases among the three most common BM: melanoma metastasizes to the cerebellar substantially less frequently than lung and breast cancers. One prior single-institution surgical series and a second radiographic series have reported similar results[37]. Taken together with this prior data, the large, multi-institutional nature of our dataset makes a statistical anomaly unlikely as an explanation. The biological reasons underlying this difference remain to be elucidated, but investigation into the microenvironmental differences between the supratentorial brain and cerebellum may help shed light on targetable adaptive mechanisms by which metastases preferentially colonize and grow in different brain compartments. Intriguingly, prior work has demonstrated differences between melanoma and lung cancer cell behavior in an in vivo model of brain metastasis initiation, with melanoma tending to maintain close contacts to microvessels and to develop perivascular growth via vessel cooption, while lung cancer metastases underwent early angiogenesis[9].

The deep structures of the brain that we demonstrate to be relatively spared from metastases are increasingly appreciated to play critical roles in neurologic function. For example, the thalamus is a complex and critical brain structure in mediating and orchestrating cortical, subcortical, and cerebellar networks. Sub-regions of the thalamus play key roles in attention filtering, the ability to focus on a target in an environment containing distractors, encoding and retrieval of memory, and arousal and awareness[38]. The internal capsule conveys a concentration of afferent and efferent fibers which connect cortical and subcortical structures, and lesions of the internal capsule can result in memory impairment[39]. Regions of the cingulate cortex are well-known as hubs of the default mode network[40], and play critical roles in information processing, directing attention, and in emotion and behavior control[41]. The orbitofrontal cortex plays an important role in executive control, learning and working memory[42]. The central importance of the hippocampi in memory has long been known, and radiation is postulated to impair hippocampal neurogenesis and disrupt the neuronal microenvironment via microvasculature injury and inflammation[43]. WBRT-HA, in which the hippocampi are constrained to receive a reduced radiation dose, combined with memantine, has

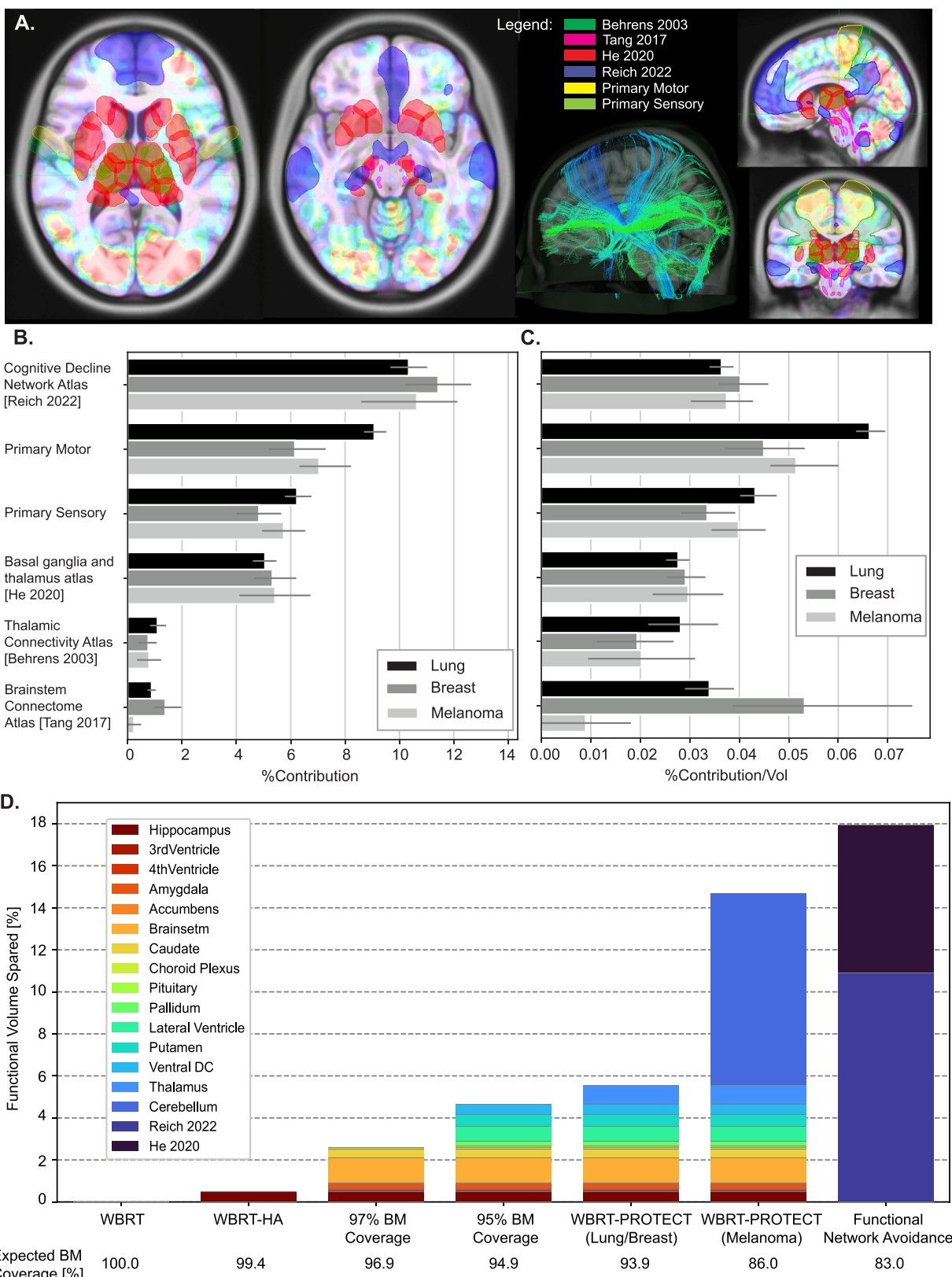

been shown in a randomized Phase III trial, NRG-CC001[18], to reduce the probability of cognitive decline. However, the magnitude of benefit was modest, and the majority of patients in both arms experienced cognitive decline after radiation.

Given the distributed nature of many cognitive functions such as memory and attention, it is likely that efforts to spare additional

structures with known importance to cognition and memory, informed by the probability of BM formation, could yield even greater sparing of cognitive decline. To this end, our proposed WBRT-PROTECT framework, based upon tailoring function-sparing radiotherapy to varying degrees of probability of undertreating potential metastatic disease and weighting radiation dose based upon

**Fig. 3 | Brain metastasis spatial distribution by functional regions and tradeoff of whole-brain RT sparing techniques and expected BM coverage. A** Brain metastasis risk levels displayed on representative axial, sagittal, and coronal slices of the MNI model for patients with lung primary cancer ($N = 1315$ patients, $L = 5230$ lesions) at first diagnosis of brain lesions prior to stereotactic treatment, as well as neuro-cognitive, motor, and sensory relevant functional maps. Specific functional atlases include: the Thalamic Connective atlas (Behrens 2003), the Brainstem Connectome atlas (Tang 2017), the Basal Ganglia and Thalamus atlas (He 2020), and the Cognitive Decline Network atlas (Reich 2022). **B** Percentage contribution of brain metastasis lesions by MNI population functional maps for lung ($N = 1315$ patients, $L = 5230$ lesions), breast ($N = 570$ patients, $L = 3007$ lesions), and melanoma ($N = 407$ patients, $L = 1931$ lesions) primary cancer. Error bars indicate 95% confidence interval over the four institutions represented in the dataset. The center of the error bars represents the mean percentage contribution of all four institutions for a given structure. **C** Percentage contribution of brain metastasis normalized by the volume by MNI functional regions for lung ($N = 1315$ patients, $L = 5230$ lesions), breast ($N = 570$ patients, $L = 3007$ lesions), and melanoma ($N = 407$ patients, $L = 1931$ lesions) primary cancer. Error bars indicate 95% confidence interval over the four institutions represented in the dataset. The center of the error bars represents the mean percentage contribution normalized by the volume of all four institutions for a given structure. **D** Examples of functional sparing treatment volumes and the associated expected BM coverage ($N = 2305$ patients, $L = 10,398$ lesions). Colored regions indicate spared regions contained within each proposed treatment. Coverage reported as the mean between all primaries, except for "Melanoma PROTECT" which is only coverage of expected Melanoma BM. Source data are provided as a Source Data file.

probabilistic maps of sites of metastasis predilection, may provide a basis for greater personalization of this critical therapy for BM patients. Moreover, the selective sparing of deep and central brain structures is particularly well suited to an application of proton therapy, taking advantage of the abrupt Bragg peak effect resulting in a lack of exit dose for this radiation modality. In case of future development of metastases within spared regions, SRS could then be used to target new lesions while maximizing preservation of cognitive function.

However, several limitations must be acknowledged. Firstly, the retrospective nature of the study may introduce biases related to historical treatment decisions and institutional imaging practices. Secondly, the patient population studied was limited to patients presenting for first radiosurgery. These biases could affect the generalizability of our findings to current clinical scenarios of re-treatment and adaptation of CNS-penetrant agents. Comparison with patients with >10 BM (i.e., more likely candidates for traditional WBRT) is necessary, particularly for evaluating the feasibility of WBRT-PROTECT. This comparison may highlight interesting differences in BM distribution between limited/early and diffuse/advanced intracranial metastatic disease that could lead to a better biologic understanding of BM development. Furthermore, while the data collection was standardized across institutions, variations in imaging protocols and radiation therapy techniques may introduce heterogeneity that complicates the interpretation of results. A notable limitation is the transformation of image datasets to a standardized space (MNI model) is subject to registration errors. Despite using the most accurate registration techniques available to minimize inaccuracies, the inherent variability in brain morphology and the presence of BM can lead to errors that may impact the spatial risk analysis. The ~5 mm accuracy in lesion centroid mapping, while acceptable, still leaves room for improvement that could potentially alter the risk distribution patterns observed. It must also be considered that the study does not account for the dynamic nature of BM. The spatial distribution of BM at the time of initial radiosurgery does not necessarily reflect the evolution of disease over time or the influence of systemic therapy. Additionally, while our study includes a wide range of primary cancers, the heterogeneity of BM from rarer primaries or those with unique molecular subtypes may not be fully represented for study at this time. Future work will focus on collecting and analyzing histopathological and molecular information on patients' cancers to correlate brain metastasis distribution with genetic and histologic variables.

The future trajectory of this research will involve rigorous, larger-scale prospective validation of the risk maps and a widened scope to encompass a greater array of clinical and BM features, with the ultimate goal of improving the lives of patients with brain metastases.

## Methods

### Study inclusion criteria and datasets

In this multi-institutional retrospective study, BM patients previously treated after initial diagnosis of BM with radiosurgery were investigated. This study was approved by the Institutional Review Board at each of the participating institutions (UCSF: IRB# 20-32526, UCD: IRB# 2005950, PMH: REB CAPCR # 17-5662 and CAPCR# 18-5368, and MSKCC: IRB# 16-1307), and written informed consent for study inclusion was obtained from patients. Data transfer agreements were created with all participating institutions. As illustrated in Fig. 1, each of the participating institutions completed a clinical data template that specified demographic data (e.g., age, gender), primary tumor location, histology, molecular subtypes, and any combined immunotherapy at the time of first radiosurgery. Patients presenting with leptomeningeal disease were excluded from the study. The study was limited to patients with comprehensive clinical and demographic data, and suitable pre-radiosurgery T1-weighted MR imaging with existing tumor segmentation. Patients who were previously treated with radiotherapy (radiosurgery or whole-brain irradiation) or surgery were excluded. The clinical data template, tumor characteristics, lesion distance to the MNI white-gray matter junction, and lesion centroid presence in the region-of-interest (ROI) atlases were consolidated into a SQL database for plotting and analysis.

### Code development

A data processing pipeline was developed to sort, assemble, and analyze DICOM RT files. The pipeline was designed entirely using open-source packages, including SQLite, OpenCV, SimpleITK, NumPy, and Pydicom. Core design principles centered around robustness, efficiency, and scalability. Robustness was ensured by designing the framework to handle many of the variations in the DICOM implementation, including the definition of structures in any orientation and structures with both connected and disconnected subspaces. Parallelization of all computationally expensive operations ensured rapid ingestion (170 studies/second) and conversion from DICOM to NIFTI (1.5 seconds/study). As implemented, the data processing pipeline can facilitate expansion of the cohort at a rate of over a thousand patients per hour on pre-existing hardware.

Upon data ingestion of the anonymized DICOM files, the pipeline stripped relevant header data fields and stored them in a SQL database. Using the database to optimize conversion, each ROI was converted to a NIFTI file. Measures were taken to ensure robustness for non-square images, and accurate alpha-shape to polygon conversion. Using these NIFTI files, a set of statistics were computed upon each individual ROI. These statistics included volume, sphericity, Euclidean distance from lesion to the white–gray matter interface, and lesion centroid location within each atlas. Results from these calculations were stored in a SQL database for streamlined access, parsing, and analysis. After the synthesis of composite risk maps, the data processing pipeline allowed for the creation of DICOM-compliant image or structure set files, allowing these files to be transferred for use in other clinical software.

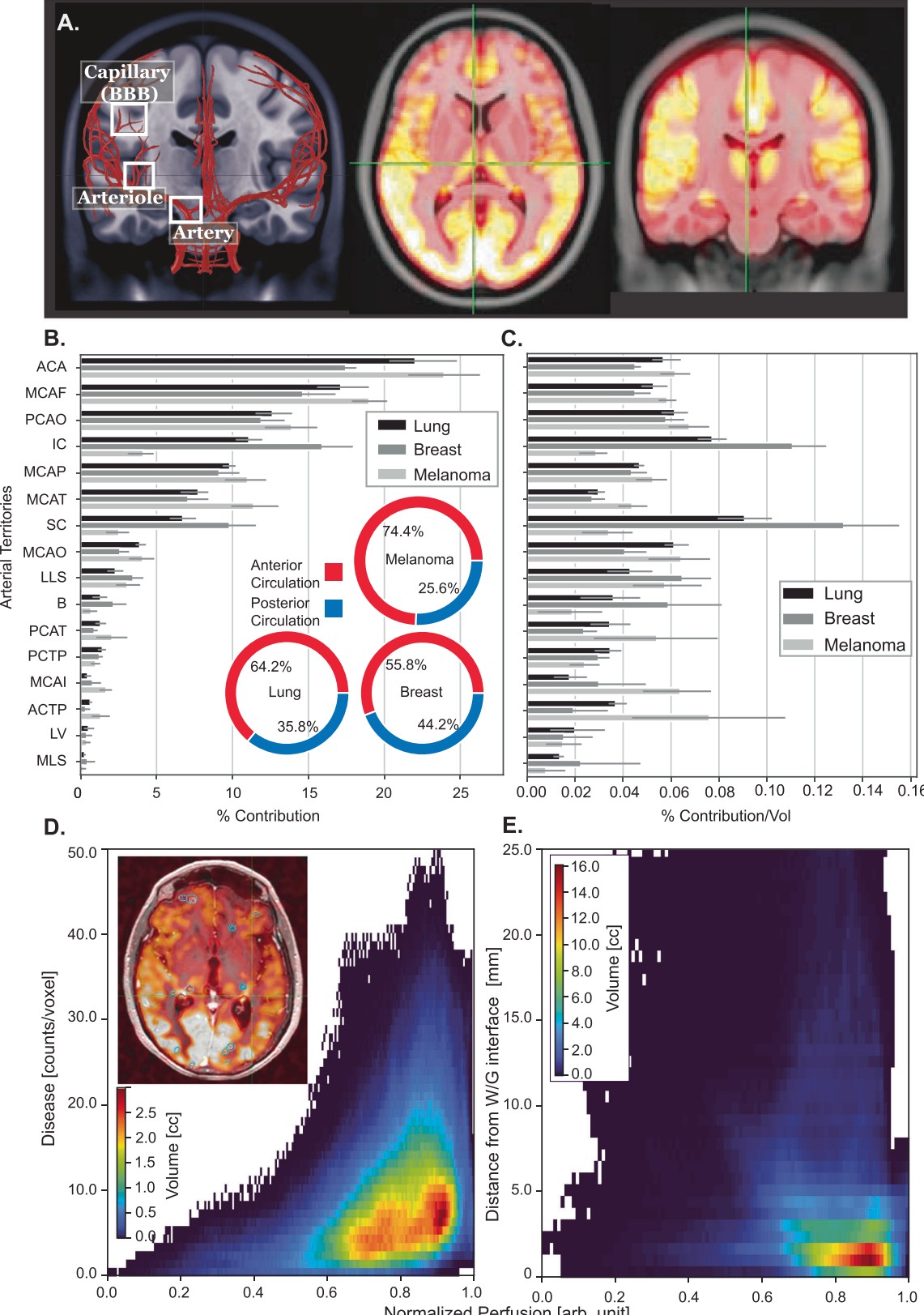

## Image processing, plotting, and analysis

Pre-radiosurgery images from all institutions were acquired across multiple vendor MR scanners with varying field strengths (either 1.5 or 3.0 Tesla). While MR imaging parameters varied slightly within each institution as well as across institutions, all patients had a pre-radiotherapy post-gadolinium contrast T1-weighted image. Post-gadolinium contrast T1-weighted MR images were used as part of patients' standard of care, where individual BM were identified and manually contoured by the attending radiation oncologists (UCSF: S.B., P.K.S., J.N., L.B., D.R.R., H.V., J.C., S.F.; UCD: R.F.; P.M.H.: D.B.S.;

**Fig. 4 | Brain metastasis spatial distribution by level of perfusion and arterial territories. A** Left- illustration of brain vascular levels: artery, arteriole, and capillary. Center/right—representative arterial spin labeling image on representative axial and coronal slices on the MNI model. **B** Percentage contribution of brain metastasis lesions by MNI population arterial territories for lung ($N$ = 1316 patients, $L$ = 5224 lesions), breast ($N$ = 565 patients, $L$ = 2998 lesions), and melanoma ($N$ = 406 patients, $L$ = 1954 lesions) primary cancer. Error bars indicate 95% confidence interval over the four institutions represented in the dataset. The center of the error bars represents the mean percentage contribution of all four institutions for a given structure. Percentage contribution of brain metastasis in sub-arterial territories combined into anterior and posterior arterial territories for lung, breast, and melanoma primaries. **C** Percentage contribution of brain metastasis normalized by the volume by MNI arterial territories for lung ($N$ = 1316 patients, $L$ = 5224 lesions), breast ($N$ = 565 patients, $L$ = 2998 lesions), and melanoma ($N$ = 406 patients, $L$ = 1954 lesions) primary cancer. Error bars indicate 95% confidence interval over

the four institutions represented in the dataset. The center of the error bars represents the mean percentage contribution normalized by the volume of all four institutions for a given structure. **D** Brain metastasis volume and count per voxel as a function of normalized brain perfusion on the MNI model. Color map indicates total brain volume within the joint histogram bin. **E** Brain metastasis volume plotted with the minimum white–gray matter distance as a function of normalized brain perfusion of the MNI model. ACA anterior cerebral artery, MCAF frontal pars of middle cerebral artery, MCAP parietal pars of middle cerebral artery, PCAO occipital pars of posterior cerebral artery, MCAT temporal pars of middle cerebral artery, IC inferior cerebellar, LLS lateral lenticulostriate, SC superior cerebellar, MCAO occipital pars of middle cerebral artery, PCAT temporal pars of posterior cerebral artery, B basilar, MCAI insular pars of the middle cerebral artery, PCTP posterior choroidal and thalamoperfurators, ACTP anterior choroidal and thalamoperfurators, MLS medial lenticulostriate, LV lateral ventricle. Source data are provided as a Source Data file.

MSKCC: L.P., J.J.N.) as part of the patient's radiotherapy treatment planning. MR images as well as their respective radiotherapy contours structures were exported from the treatment planning system to a local archive, de-identified (using safe harbor) per the DICOM standard and imported in a commercially available image processing software (MIM Software Inc., Cleveland, OH).

MR images were bias corrected[44] and normalized in MIM to limit variability in signal intensity between images. Each individual MR image and contour structure set was registered to the MNI standard[45,46], first using a rigid affine registration, followed by a deformable registration available in MIM. This mapping methodology was previously described in multiple studies[37,47–49], and shown to improve image registration accuracy between the stationary and moving images. A review by a medical physicist (O.M.) of 300 randomly selected unique patients (-10% of the overall cohort) from the database indicated that this mapping of lesion centroids (from patient coordinate to the MNI model) achieved an accuracy of -5 mm.

All registered datasets were exported from MIM and analyzed in Python. Mathematically, all BM lesions $l$ from historical patients $i$ were accumulated on the MNI model voxels $v$ to create a cumulative count in the MNI space $C_{MNI}(\vec{x}_{MNI})$ as:

$$C_{MNI}(\vec{x}_{MNI}) = \sum_{i=1}^{N} \sum_{l=1}^{L} \sum_{v=1}^{V} \left[ c_{ilv}(\vec{x}_i) \right]_{T_i} \text{[counts per voxel]} \quad (1)$$

where $C_{MNI} \in \mathbb{R}\left[0 - C_{max}\right]$ and $C_{max}$ is the maximum lesion count from all voxels after the accumulation of all patients' lesions in the MNI space. $c_{ilv}(\vec{x}_i)$ is the binary classification (tumor = 1, no tumor = 0) of all voxels $v$ in the MNI space as transformed using $T_i$ (transform from the patient space to the MNI space). $T_i$ represents two consecutive image transformations: (1) rigid body registration, followed by a (2) deformable registration.

The derived BM count distribution $C_{MNI}(\vec{x}_{MNI})$ on the MNI brain was investigated as a function of anatomy, functional tracks, and brain perfusion. Previously published MNI brain segmentation and FastSurfer[50] for sub-anatomy, yielded 145 ROIs used to derive brain metastases count within each ROI. Functionally, the Human Connectome Project tracks[19,20,22,51,52] were used for risk analysis. Additionally, brain perfusion atlases were derived using two distinct approaches. The first being a previously derived digital 3D brain MRI atrial territories atlas[53] and the second a composite of 50 high-quality 3 T 3D MRI arterial spin labeled (ASL) scans from UCSF patients registered to the MNI brain.

Supplemental Fig. 2 was generated to assess if the observed over-representation of BMs in proximity to the white-gray matter interface is due to physiology or an artifact of the convoluted morphology. For

each institution, the BM occurrence and total volume (null histogram) was computed at 0.5 mm intervals from the white-gray matter interface. The BM occurrence and null histogram were compared using the Wasserstein distance (Earth mover's distance) and Wilcoxon-signed rank test, with a significance of $P$ value < 0.05.

## Statistical analysis

Statistical analysis was performed in Python to establish the significance of observed differences in the BM risk maps (percentage lesion count per ROI of the anatomy, functional, and vascular atlases) between primaries and other factors. Error bars indicate a 95% confidence interval over the four institutions represented in the study. All $P$ values computation was performed using the two-tailed Welch's $t$ test and considered significant if $P$ value < 0.05.

## Treatment planning

The MNI phantom with associated contour atlases (anatomical coarse, anatomical fine, functional, and vascular) was imported into a commercial radiation planning software (RayStation V9.0, RaySearch Laboratories, Stockholm, Sweden) used in the clinic for photon and proton treatment simulations. The external surface of the MNI phantom was contoured with density and stopping power overridden to water equivalent tissue. Four plans were generated to represent the different whole-brain RT approaches. The conventional planning techniques included WBRT delivered with two opposed lateral photon beams per standard of care and WBRT-HA planned using volumetric modulated arc therapy (VMAT) delivery to achieve the NRG-CC001 protocol tolerances to the hippocampi and 5 mm surrounding region. For the treatment proposal, WBRT-PROTECT, plans were generated for the most commonly used photon linear accelerator in North America (Varian TrueBeam; Varian Medical Systems, Palo Alto, CA) and using proton therapy.

The beam arrangement used for photon treatment planning was designed to replicate the Varian HyperArc setup with two coplanar arcs, two 45-degree couch rotations, and a vertex beam. Treatment plans were generated using the six MV clinical beam model for a Varian TrueBeam with the Millennium 120 multi-leaf collimator. Treatment arc control points were limited to every 2 degrees, and the maximum dose rate was limited to 600 MU/second. An optimization tolerance of 1e-7 was used with an intermediate dose calculation performed at 45 iterations, and a maximum of 85 iterations. After each optimization, a final dose calculation was performed to a 2 mm³ dose grid using a collapsed cone convolution algorithm. Optimization objectives were selected to minimize the mean dose to the PROTECT region while

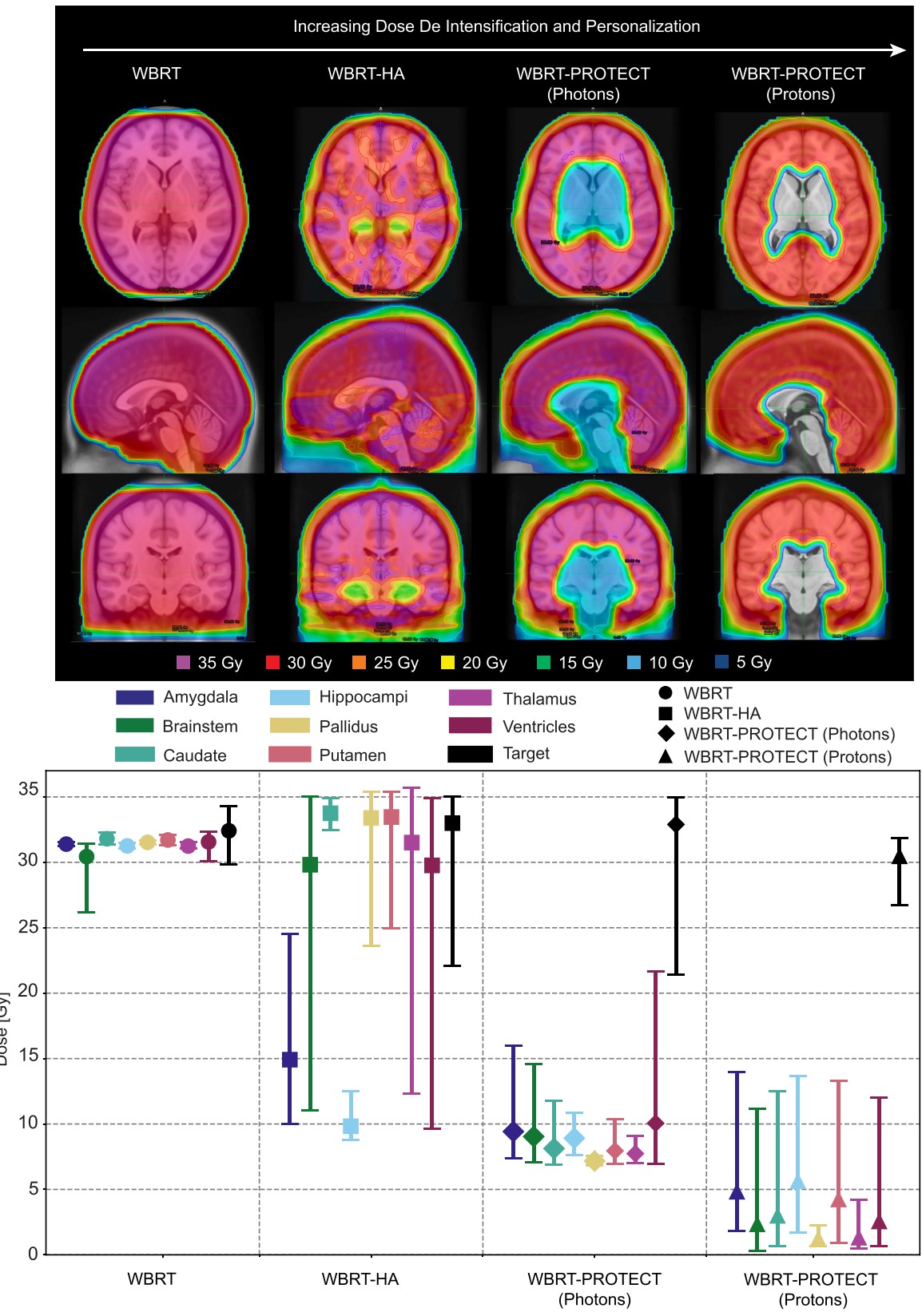

maintaining coverage of at least 95% of the WBRT-PROTECT treatment volume at 30 Gy. Dose fall−off was controlled between the prescription dose to 12.5 Gy to within 0.5 cm, with additional control on dose fall−off into the optic structures.

For the intensity-modulated proton therapy (IMPT) treatment planning, proton doses were specified in Gray (Gy(RBE)), utilizing a consistent relative biological effectiveness (RBE) factor of 1.1. Inverse treatment planning was performed using a single

**Fig. 5 | Comparison of WBRT techniques, dose distributions, and functional sparing potential.** Top−treatment plans ordered by progressing dose de-intensification on representative axial, sagittal, and coronal slices. From left to right: the conventional techniques of WBRT, WBRT-HA, the newly proposed personalized approaches with higher functional sparing PROTECT (Personalized Radiation Optimization To Eliminate Collateral Toxicity) techniques planned using photons WBRT-PROTECT(Photons) or protons WBRT-PROTECT(Protons).

Bottom−error bars represent the minimum, mean and maximum dose metrics achieved to compares the target[black] and selected functional structures (hippocampi [light blue], amygdala [dark blue], brainstem [dark green], putamen [pink], caudate [light green], pallidus [yellow], thalamus [purple] and ventricles [dark purple]) for the WBRT[circles], WBRT-HA[squares], WBRT-PROTECT(Photons)[diamonds], and WBRT-PROTECT(Protons)[triangles] plans. Source data are provided as a Source Data file.

field optimization option with simultaneous spot optimization, where the spot position, energy, and number of monitor units (MUs) were determined by treatment planning system incorporating constraints for target volumes and critical structures. A total of 8 beams (7 coplanar and 1 vertex) were utilized to generate the WBRT-PROTECT-PROTON plan. During the optimization process, energy layer spacing and spot spacing were automatically set with a scale of 1.0, and the spot pattern was set to hexagonal. The tolerance was set at 1e−5, allowing for up to 100 iterations to refine the treatment plan. The minimum spot meterset was set as 0.01 MU/fraction, while the maximum spot meterset was set to the machine default of 15 MU/fraction. Robustness settings included an isotropic uncertainty of 0.3 cm in all directions, and the systematic density uncertainty was set at 3.5%. Optimization objectives and constraints were strategically chosen to ensure the target volume received both the minimum and maximum prescribed doses. The dose fall−off of the protected neuro structure had a high dose level set to the prescription dose, with a low dose level of 1000 cGy (RBE) at a distance of 0.5 cm.

The WBRT-PROTECT target and neuro-protect volume were generated using ROI Boolean operations. The neuroPROTECT volume was created from a 3 mm isotropic expansion of the union of the hippocampi, amygdala, thalamus, pallidus, caudate, putamen, brainstem, and ventricles. Both WBRT-PROTECT plans used a target volume defined as the whole-brain minus the neuroPROTECT region. All plans were prescribed using the standard of care of 30 Gy in 10 fractions to 95% coverage of their respective planning target volume. Dose distributions with the MNI image and contours were exported from RayStation to Python for plotting of dose-volume histograms and extraction of dose metrics of the plans.

### Reporting summary
Further information on research design is available in the Nature Portfolio Reporting Summary linked to this article.

## Data availability
The dataset supporting this study includes multi-institutional data subject to privacy regulations and institutional data use policies. As such, full access is restricted. Researchers may request access to a limited, de-identified subset of the data derived from the University of California, San Francisco (UCSF) under a controlled-access Material Transfer Agreement (MTA). Requests should be directed to the corresponding author. All requests will be reviewed within 30 days. Data use is subject to restrictions outlined in the MTA, including limitations on redistribution and requirements for secure storage. To promote transparency and reproducibility, a curated test dataset of 500 anonymized patients, along with analytical code, is publicly available via the Open Science Framework (OSF): https://osf.io/fkqmr/. Additionally, we provide open access to the MNI brain model, anatomical, functional, and vascular atlases, brain metastasis risk maps (lung, breast, and melanoma), and simulated radiation treatment models. Source data are provided with this paper.

## Code availability
Code is directly available from the following GitHub repository: https://github.com/medomics/BrainMets_RiskMaps: https://doi.org/

10.5281/zenodo.15132555: Alternatively, code is available via the Open Science Framework (OSF) repository: https://osf.io/fkqmr/.

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

## Acknowledgements

We would like to thank patients who signed the institutional consent form such that their de-identified care data can one day benefit others.

## Author contributions

Conceived and designed the overall study, methodology, interpreted the data, and writing—review and editing: J.B., E.P., D.C., T.U., W.C., J.R.P., A.A., A.B., K.P., M.A., E.L., D.H., P.H.S., J.E.V.M., G.V., F.J., O.W.C., B.Z., M.M., H.L., M.S., P.S., J.N., L.B., S.F., D.R., J.C., H.V., S.C., C.H., R.F., D.B.S., L.P., S.H.J., P.T., J.Y.S., M.A.K.A., D.C., S.H.B., K.S., J.S., C.C., D.S.T., J.D., S.B. and O.M. Obtained data access and IRB approval, and patient selection and data curation: M.A., C.C., J.R.P. and O.M. Creation of the brain mets SQL database: J.B. and E.P. Funding acquisition and funding sources: not applicable. Project administration: O.M., C.C., J.D. and S.H.B. Website maintenance: O.M. Software and designed and constructed the predictive modelling: J.B., E.P., D.C., T.U., W.C. and O.M. Writing—original draft: J.B., E.P., W.C., D.C., J.E.V.M., S.B and O.M.

## Competing interests

The authors declare no competing interests.

## Additional information

[1]Department of Radiation Oncology, University of California San Francisco, San Francisco, CA, USA. [2]Department of Radiation Oncology, University of California Davis Health, Sacramento, CA, USA. [3]Department of Radiation Oncology, Memorial Sloan Kettering, New York, NY, USA. [4]Department of Medical Physics, Memorial Sloan Kettering Cancer Center, New York, NY 10065, USA. [5]Department of Radiology and Biomedical Imaging, University of California San Francisco, San Francisco, CA, USA. [6]Department of Neurological Surgery, University of California San Francisco, San Francisco, CA, USA. [7]Department of Epidemiology and Biostatistics, University of California San Francisco, San Francisco, CA, USA. [8]Moffit Cancer Center, Tampa, Florida, USA. [9]Radiation Medicine Program, Princess Margaret Cancer Centre, University Health Network, Toronto, ON, Canada. [10]Department of Medical Biophysics, University of Toronto, Toronto, ON, Canada. [11]UCSF/UC Berkeley Graduate Program in Bioengineering, San Francisco, CA, USA. [12]Department of Pathology, University of California San Francisco, San Francisco, CA, USA. [13]These authors contributed equally: Jorge Barrios, Evan Porter. ✉e-mail: Olivier.Morin@ucsf.edu

