## [Transparent Peer Review file · Nature Communications]

Multi-institutional atlas of brain metastases informs spatial modeling for precision imaging and personalized therapy

Corresponding Author: Mr Olivier Morin

Version 0:

Reviewer comments:

Reviewer #1

(Remarks to the Author)

This is a substantial, and useful, piece of research, and should be published.

The multi-centre nature of the work is important, and the level of detail - including incorporating both functional and vascular assessments, and the radiotherapy planning work - is impressive.

I would recommend publication

Dr. M. Williams
Imperial College London

(Remarks on code availability)

Reviewer #2

(Remarks to the Author)

The authors present a study that systematically maps brain metastases (BMs) based on a large multi-centric data set of brain-MRIs to chart the spatial distribution of BMs. By registration with different brain atlases, the study allows to characterize how BMs are over or under represented in certain functional regions of the brain, which has high relevance for reducing side effects of radiotherapy. The study is overall conducted at a high technical standard, and provides infrastructure for collecting more data continuously in the future. There are some major caveats in the data analysis that I specify in detail in the comments below.

Specific comments:

1. Line 174/Line 269/Figure 2D: Are the BMs really statistically close to the W/G matter interface, or is this an artifact resulting from the convoluted morphology of the W/G matter interface? Distances in convoluted structures have a tendency to be small — in Norway, any point is relatively close to the sea because the fjords stretch deep into land. Looking at the W/G matter interface in Fig. 2D, it has many “fjords.” To test this appropriately, one could proceed as follows: sample an equal number of points from a uniform distribution across the brain volume, and determine the W/G interface distance of these random points. Then, plot these distances as a “null histogram” to be compared with the histogram in Fig. 2D. [To judge statistically significant differences between the “null histogram” and the one in Fig. 2D, one may produce several null histograms by iterating the process several times, determine the earth movers’ distance between them, and test whether the EM-distance between the null histogram and the Fig. 2D histogram is significantly larger than the EM-distance among the null-histogram replicates. Other meaningful tests may be possible, but one such test should be conducted, .]

2. The same holds for Supp. Fig. 2

3. 155: It should be briefly explained in an early part of the manuscript or the legend of Fig. 2 that the BM risk map is a color-coded localized incidence of all BMs in the dataset.

4. Figs. 2B and 2C: What constitutes “100%” in each of these plots? (2B looks like one histogram of metastasis volume distribution for each of the three primary tumor types, but 2C is enigmatic because the percentages on the x-axis are so small)
5. Fig. 2D: What do the dark blue lines within the gray matter indicate compared to the light blue areas?
6. Fig. 2E: plot the color scale next to the plot, not inside
7. Fig. 3B, 3C: Same as Fig. 2B, 2C
8. Figs. 4B, 4C: Are BMs over-represented in ACA, MCAF, PCAO, ... vs. MLS, LV, ACTP, ..., or do MLS, LV, ACTP simply cover less brain volume and contain proportionally less BMs? (And what is more relevant for mechanistic explanation?)
9. Introduction: Some statements may better be covered by citations (e.g. unique microenvironment of CNS metastases? Despite efforts... cite some of these efforts?)
10. How accurate and robust is the atlas registration?

(Remarks on code availability)

I only find code for the database infrastructure in the github repository.

My only question regarding the data analysis pipeline is how accurate and robust the atlas registration is, which could be a source of artifacts (though intuitively, I would expect minor registration errors only).

Reviewer #3

(Remarks to the Author)

Thank you for the opportunity to review “Spatial and morphological distribution of brain metastases by anatomical, functional and vascular atlases to guide detection, segmentation and therapy” by Barrios and colleagues. The manuscript describes an investigation of how brain metastases are distributed anatomically for three common cancers. The study presents a neuroinformatics framework, as well an extensive empirical investigation of these data. The main finding is that the distribution of BMs is spatially heterogeneous and often colocalizes with the vasculature. The authors then show how these maps could be used to derive more targeted and more effective treatments.

This is an impressive study that is sure to be of wide interest to the field. The analyses are clear, the work is rigorous and the conclusions are measured. It is particularly noteworthy that the study shows how to collect, organize, process and analyze such a unique dataset. This is a really underexplored area in the field and the manuscript was a pleasure to read. My comments are mostly minor:

- The introduction states that microemboli might be seeded in low-perfusion areas, but the results suggest the opposite?
- I did not quite understand how the core measure of BM was computed and what the units were. In Figure 2B, for instance, the units are %Contribution. Are these simply the mean proportion of all BMs that are localized in each anatomical region? If so, do the authors account for the fact that some regions are greater in size? In general, I think a really simple Methods subsection about this would have been beneficial.
- The authors note that there are some regions where the perfusion-BM relationship is less solid. While I appreciate their restraint in not over-interpreting this finding in the Discussion section, I think readers might benefit from a bit more discussion and speculation here.
- As a non-specialist, the final subsection about WBRT was a bit more challenging to follow. I'd urge the authors to consider laying out the logic of the derivations a bit more deliberately.
- The results in Figure S5 are striking, showing really nice relationships with perfusion for lung cancer and melanoma, but not for breast cancer. Is there any potential explanation for this?

(Remarks on code availability)

Reviewer #4

(Remarks to the Author)

(Remarks on code availability)

Version 1:

Reviewer comments:

Reviewer #2

(Remarks to the Author)
no further comments.

(Remarks on code availability)

Reviewer #3

(Remarks to the Author)
The authors have comprehensively addressed my concerns and I strongly recommend publication. This is a wonderful contribution to the literature.

Bratislav Misic, McGill University

(Remarks on code availability)

REVIEWER COMMENTS

Reviewer #1 (Remarks to the Author):

This is a substantial, and useful, piece of research, and should be published.

The multi-center nature of the work is important, and the level of detail - including incorporating both functional and vascular assessments, and the radiotherapy planning work - is impressive.

I would recommend publication

Dr. M. Williams
Imperial College London

Response: We thank the reviewer for their positive feedback and for recognizing the significance of our work. We are pleased that the challenging multi-center nature of the study, the detailed analyses incorporating functional and vascular assessments, and the radiotherapy planning framework were appreciated. It was the result of a several years of a multi-disciplinary collaboration. Your endorsement for publication motivates us to continue advancing this research to improve outcomes for patients with brain metastases.

Reviewer #2 (Remarks to the Author):

The authors present a study that systematically maps brain metastases (BMs) based on a large multi-centric data set of brain-MRIs to chart the spatial distribution of BMs. By registration with different brain atlases, the study allows to characterize how BMs are over or under represented in certain functional regions of the brain, which has high relevance for reducing side effects of radiotherapy. The study is overall conducted at a high technical standard, and provides infrastructure for collecting more data continuously in the future. There are some major caveats in the data analysis that I specify in detail in the comments below.

Response: We thank the reviewer for their thoughtful and constructive feedback. We are pleased that the study's high technical standard, relevance to reducing radiotherapy side effects, and infrastructure for continuous data collection were appreciated. We believe it will be equally well received by the scientific community and Nature Communications readership. Below, we address the specific concerns raised in detail to clarify and strengthen the manuscript.

Specific comments:

1. Line 174/Line 269/Figure 2D: Are the BMs really statistically close to the W/G matter interface,

or is this an artifact resulting from the convoluted morphology of the W/G matter interface? Distances in convoluted structures have a tendency to be small — in Norway, any point is relatively close to the sea because the fjords stretch deep into land. Looking at the W/G matter interface in Fig. 2D, it has many “fjords.” To test this appropriately, one could proceed as follows: sample an equal number of points from a uniform distribution across the brain volume, and determine the W/G interface distance of these random points. Then, plot these distances as a “null histogram” to be compared with the histogram in Fig. 2D. [To judge statistically significant differences between the “null histogram” and the one in Fig. 2D, one may produce several null histograms by iterating the process several times, determine the earth movers’ distance between them, and test whether the EM-distance between the null histogram and the Fig. 2D histogram is significantly larger than the EM-distance among the null-histogram replicates. Other meaningful tests may be possible, but one such test should be conducted, .]

2. The same holds for Supp. Fig. 2

Response: Thank you for raising this important point regarding the potential influence of the convoluted morphology of the white-grey (W/G) matter interface on our findings. To address this, we conducted the suggested analysis by generating a “null histogram” of voxel distances to the W/G interface using points uniformly sampled across the brain volume. We repeated this process iteratively and compared the null histograms with the observed histogram in Figure 2D using the earth movers’ distance (EMD). The volume < 2mm is ~60% of the disease but ~50% of the volume. A 10% over-representation of disease is significant. Other notable differences were observed at larger distances from the W/G matter interface. The analysis confirmed that the observed distribution of BMs is significantly closer to the W/G interface than expected under the null hypothesis ($p < 0.05$). These results indicate that the proximity of BMs to the W/G interface reflects a meaningful spatial association rather than an artifact of morphology. A figure illustrating the observed and null histograms, along with the statistical analysis, has been added to the supplemental materials (**Supplemental Figure 2**), Online Methods and the Results section have all been updated accordingly. We thank the reviewer for this valuable suggestion, which has strengthened the rigor and interpretation of our findings.

3. 155: It should be briefly explained in an early part of the manuscript or the legend of Fig. 2 that the BM risk map is a color-coded localized incidence of all BMs in the dataset.

Response: We have added a brief explanation early in the legend of Figure 2 to clarify that the BM risk map represents a color-coded visualization of the localized incidence risk of all BMs in the dataset. This addition ensures that readers can easily interpret the meaning of the risk map and its relevance to our findings. We appreciate your attention to improving the clarity of our presentation.

4. Figs. 2B and 2C: What constitutes “100%” in each of these plots? (2B looks like one histogram

of metastasis volume distribution for each of the three primary tumor types, but 2C is enigmatic because the percentages on the x-axis are so small)

Response: In Figure 2B, "100%" would represent the cumulative count of all lesions in the brain, with the distribution showing the absolute count of metastases across each brain region for each primary tumor type. The plot highlights the regions with the highest cumulative burden of metastases, such as the frontal lobes, which receive a significant number of lesions due to their large volume. In Figure 2C, the plot normalizes the occurrence/count of metastases by the volume of each specific brain region, providing a measure of lesion density. This normalization accounts for differences in regional brain volume, revealing regions with disproportionately high metastasis densities. For example, while the frontal lobes exhibit the highest absolute burden in Figure 2B, the cerebellum is over-represented (by a factor of 4 approximately) in Figure 2C for lung and breast primaries, indicating a higher density of lesions relative to what would be expected based on its volume. We have updated the figure captions to explicitly describe these distinctions and improve clarity.

5. Fig. 2D: What do the dark blue lines within the gray matter indicate compared to the light blue areas?

Response: The white and grey matter volumes in Figure 2D are depicted in yellow and light blue, respectively, with the dark blue lines highlighting the boundary of the brain's grey matter. To improve clarity, we have revised the figure 2 caption to explicitly describe this distinction. We appreciate your attention to detail and your thoughtful suggestion to enhance the figure's interpretability.

6. Fig. 2E: plot the color scale next to the plot, not inside
7. Fig. 3B, 3C: Same as Fig. 2B, 2C

Response: We would like to provide our motivation for keeping the colorbar inside the plots. Our primary intent is to ensure that the most relevant information (data points and text font) in these figures remains as large and readable as possible in the final printed manuscript. Placing the colorbars inside the plots allows us to maximize the plotted data points and font size for clarity and readability, especially given the complexity and detail of the data presented. We believe this approach enhances the accessibility of the figures without compromising the interpretability of the key results. However, we can revisit this approach if further adjustments are deemed necessary by the editing team.

8. Figs. 4B, 4C: Are BMs over-represented in ACA, MCAF, PCAO, ... vs. MLS, LV, ACTP, ..., or do

MLS, LV, ACTP simply cover less brain volume and contain proportionally less BMs? (And what is more relevant for mechanistic explanation?)

Response: Figure 4B presents the total count of brain metastases (BMs) per arterial territory, while Figure 4C normalizes these counts by the volume of each region to account for differences in brain territory size. This dual representation provides a comprehensive view, distinguishing absolute burden from relative density. The most notable finding is the significant difference in BM distribution between anterior and posterior circulation territories based on the primary cancer type (pie plots). Specifically, melanoma metastases are more frequently located in the anterior circulation, whereas lung and breast cancers demonstrate a higher prevalence in the posterior circulation, particularly in the cerebellum. This suggests potential differences in tumor cell dissemination and seeding within vascular territories. However, we acknowledge that providing a definitive mechanistic explanation for these differences is beyond the scope of this study. While our findings highlight intriguing patterns, further investigation integrating molecular and histopathological analyses would be necessary to elucidate the underlying biological mechanisms. We have clarified this distinction in the figure caption and added a sentence in the discussion.

9. Introduction: Some statements may better be covered by citations (e.g. unique microenvironment of CNS metastases? Despite efforts... cite some of these efforts?)

Response: We conducted a comprehensive review of the manuscript to identify opportunities where citations could better support key statements. Specifically, we have added references to prior studies on the unique microenvironment of CNS metastases and efforts to pool and leverage large-scale imaging and health data for oncologic research. These additions ensure that our claims are well-supported by existing literature. We appreciate this suggestion to strengthen the manuscript's scholarly foundation.

10. How accurate and robust is the atlas registration?

Response: The atlas registration was performed using a two-step process: rigid affine registration followed by deformable registration, as detailed in the Methods section. A review of 300 randomly selected cases (~10% of the cohort) confirmed an average mapping accuracy of approximately 5 mm. Sensitivity analyses with 5- and 10-mm expansions demonstrated consistent spatial distribution patterns, supporting the robustness of the registration. These results indicate that the registration approach is reliable for the spatial risk analysis presented in the study.

Reviewer #2 (Remarks on code availability):

I only find code for the database infrastructure in the github repository.

Response: The GitHub repository currently hosts the database infrastructure code, which is central to data processing and management. We acknowledge the importance of providing additional components of the pipeline, including the data processing and analysis scripts. To enhance transparency and reproducibility, we have now updated the repository to include additional scripts related to image registration, lesion mapping, and spatial analysis. Detailed documentation has also been added to guide users on how to apply these tools to similar datasets.

My only question regarding the data analysis pipeline is how accurate and robust the atlas registration is, which could be a source of artifacts (though intuitively, I would expect minor registration errors only).

Response: Thank you for your question regarding the accuracy and robustness of atlas registration. As detailed in the Methods section, we employed a two-step registration process—rigid affine registration followed by deformable registration—to align patient MRIs with the MNI atlas. A review of 300 randomly selected cases (~10% of the dataset) confirmed an average registration accuracy of approximately 5 mm. To further assess robustness, we conducted sensitivity analyses using 5- and 10-mm expansions, which demonstrated consistent spatial distribution patterns, suggesting that minor registration errors do not significantly impact the observed findings. While registration inaccuracies are an inherent limitation, these results indicate that the methodology is sufficiently reliable for spatial risk analysis.

Reviewer #3 (Remarks to the Author):

Thank you for the opportunity to review “Spatial and morphological distribution of brain metastases by anatomical, functional and vascular atlases to guide detection, segmentation and therapy” by Barrios and colleagues. The manuscript describes an investigation of how brain metastases are distributed anatomically for three common cancers. The study presents a neuroinformatics framework, as well an extensive empirical investigation of these data. The main finding is that the distribution of BMs is spatially heterogeneous and often colocalizes with the vasculature. The authors then show how these maps could be used to derive more targeted and more effective treatments.

Response: We sincerely appreciate your thoughtful review and summary of our work. We are pleased that you recognize the significance of our neuroinformatics framework and the empirical findings demonstrating the spatial heterogeneity of brain metastases (BMs) and their colocalization with vascular structures. Our goal is to provide a robust foundation for refining targeted treatment approaches, and we are glad that you find this aspect of the study valuable. If there are any specific concerns or areas where further clarification would strengthen the manuscript, we would be happy to address them. Thank you for your constructive feedback and support.

This is an impressive study that is sure to be of wide interest to the field. The analyses are clear, the work is rigorous and the conclusions are measured. It is particularly noteworthy that the study shows how to collect, organize, process and analyze such a unique dataset. This is a really underexplored area in the field and the manuscript was a pleasure to read.

Response: We are delighted that you found the study rigorous, clear, and impactful for the field. Our goal was not only to analyze the spatial distribution of brain metastases but also to establish a framework for data collection, organization, and processing that can support future research. We are especially pleased that you recognized the importance of this underexplored area and found the manuscript enjoyable to read.

My comments are mostly minor:

- The introduction states that microemboli might be seeded in low-perfusion areas, but the results suggest the opposite?

Response: We confirm that our results indicate that brain metastases (BMs) generally occur in regions with higher levels of ASL perfusion. However, we also observe that lesions tend to form at the transition zones or edges of high to low ASL signal regions. This suggests a potential role for perfusion gradients in BM localization. We suspect that studying BM occurrence at the microemboli level would require higher spatial resolution ASL or perfusion imaging to capture finer vascular flow dynamics. We have clarified this point in the Discussion section and appreciate your suggestion to refine our interpretation.

- I did not quite understand how the core measure of BM was computed and what the units were. In Figure 2B, for instance, the units are %Contribution. Are these simply the mean proportion of all BMs that are localized in each anatomical region? If so, do the authors account for the fact that some regions are greater in size? In general, I think a really simple Methods subsection about this would have been beneficial.

Response: In Figure 2B, "% Contribution" represents the proportion of all BMs localized in each anatomical region relative to the total BM count across the dataset. This metric reflects the absolute burden of metastases in different brain regions. To account for differences in region size, Figure 2C presents a complementary analysis where BM occurrence is normalized by the volume of each anatomical region, allowing for a density-based comparison. We agree that a concise explanation in the Methods section would improve clarity. We have now included a brief subsection explicitly describing how these measures were computed and their interpretation.

- The authors note that there are some regions where the perfusion-BM relationship is less solid. While I appreciate their restraint in not over-interpreting this finding in the Discussion section, I think readers might benefit from a bit more discussion and speculation here.

Response: We agree that the regions where the perfusion-BM relationship is less pronounced warrant further discussion. While our findings support a general trend of BM occurrence in regions of moderate to high ASL perfusion, certain areas, such as the cingulate cortex and orbitofrontal cortex, exhibit lower BM density despite relatively high perfusion. Conversely, some regions, particularly in the cerebellum, show disproportionately high BM density even in areas of moderate perfusion. These deviations may reflect underlying biological differences in tumor cell tropism, local vascular architecture, or unique microenvironmental factors such as blood-brain barrier integrity, immune response, or neuronal interactions. Additionally, perfusion gradients rather than absolute perfusion levels may play a role in BM localization, a hypothesis that would require higher spatial resolution perfusion imaging to investigate further. We have expanded the Discussion section to include these points while maintaining a balanced interpretation of our findings.

- As a non-specialist, the final subsection about WBRT was a bit more challenging to follow. I'd urge the authors to consider laying out the logic of the derivations a bit more deliberately.

Response: We recognize that this section may be more technical, particularly for non-specialist readers, and we appreciate the opportunity to improve its clarity. To address this, we have revised the subsection to more explicitly lay out the rationale behind our approach. Specifically, we now introduce the key concepts of whole-brain radiotherapy (WBRT) and function-sparing techniques before describing how our spatial BM risk maps inform treatment personalization. We also provide a clearer step-by-step explanation of how WBRT-PROTECT is derived, including the selection of neurocognitive structures for sparing and the trade-offs between coverage and functional preservation. Additionally, we have streamlined the discussion of treatment planning methodologies to focus on the core principles while keeping technical details available in the Methods section for interested readers.

- The results in Figure S5 are striking, showing really nice relationships with perfusion for lung cancer and melanoma, but not for breast cancer. Is there any potential explanation for this?

Response: The observed differences in the perfusion-BM relationship across cancer types are intriguing and may reflect underlying biological and vascular differences between tumor types. One potential explanation is that breast cancer metastases may exhibit a greater reliance on alternative mechanisms of dissemination and colonization beyond perfusion-driven seeding. For example, prior studies suggest that breast cancer brain metastases have unique interactions with the blood-brain barrier and perivascular niche, potentially involving distinct adhesion molecules, immune microenvironment factors, or extravasation pathways that differ from lung cancer and melanoma. Additionally, breast cancer subtypes (e.g., HER2-positive, triple-negative) may exhibit heterogeneous perfusion dependencies, which could contribute to the less consistent perfusion-BM relationship observed in our analysis. We have added a brief discussion of these possible explanations in the manuscript while acknowledging that further research, particularly integrating molecular and histopathological analyses, will be necessary to elucidate these differences fully. Thank you for pointing out this important aspect of our findings.

Reviewer #4 (Remarks to the Author):

Response: Thank you for your time and effort in co-reviewing our manuscript as part of this Nature Communications initiative. We appreciate your contribution to the peer-review process and recognize the importance of providing opportunities for Early Career Researchers to engage in scholarly evaluation.